



# Projections of hydrofluorocarbon (HFC) emissions and the resulting global warming based on recent trends in observed abundances and current policies

Guus J.M. Velders[1,2], John S. Daniel[3], Stephen A. Montzka[4], Isaac Vimont[4,5], Matthew Rigby[6], Paul B. Krummel[7], Jens Muhle[8], Simon O'Doherty[6], Ronald G. Prinn[9], Ray F. Weiss[8], and Dickon Young[6]

[1]National Institute for Public Health and the Environment (RIVM), PO Box 1, 3720 BA
Bilthoven, the Netherlands

[2]Institute for Marine and Atmospheric Research Utrecht (IMAU), Utrecht University, Utrecht, the Netherlands

[3]Chemical Sciences Laboratory, National Oceanic and Atmospheric Administration, Boulder, Colorado, USA

[4]Global Monitoring Laboratory, National Oceanic and Atmospheric Administration, Boulder, Colorado, USA

[5]Cooperative Institute for Research in Environmental Sciences, University of Colorado, Boulder, Colorado, USA

[6]School of Chemistry, University of Bristol, Bristol, UK

[7]Climate Science Centre, CSIRO Oceans and Atmosphere, Aspendale, Victoria, Australia

[8]Scripps Institution of Oceanography, University of California San Diego, La Jolla, California, USA

[9]Center for Global Change Science, Massachusetts Institute of Technology, Cambridge, Massachusetts, USA

*Correspondence to*: Guus J.M. Velders (guus.velders@rivm.nl)





## Abstract

The emissions of hydrofluorocarbons (HFCs) have increased significantly in the past two
decades, primarily as a result of the phaseout of ozone depleting substances under the
Montreal Protocol and the use of HFCs as their replacements. Projections from 2015 showed
large increases in HFC use and emissions in this century in the absence of regulations,
contributing up to 0.5 °C to global surface warming by 2100. In 2019, the Kigali Amendment

to the Montreal Protocol came into force with the goal of limiting the use of HFCs globally,
and currently, regulations to limit the use of HFCs are in effect in several countries. Here, we
analyze trends in HFC emissions inferred from observations of atmospheric abundances and
compare them with previous projections. Total $CO_2$-eq inferred HFC emissions continue to
increase through 2019 (to about 0.8 $GtCO_2$-eq $yr^{-1}$) but are about 20% lower than previously

projected for 2017-2019, mainly because of lower global emissions of HFC-143a. This
indicates that HFCs are used much less in industrial and commercial refrigeration (ICR)
applications than previously projected. This is supported by data reported by the developed
countries and lower reported consumption of HFC-143a in China. Because this time-period
preceded the beginning of the Kigali controls, this reduction cannot be linked directly to the

provisions of the Kigali Amendment. However, it could indicate that companies transitioned
away from the HFC-143a with its high global warming potential (GWP) for ICR applications,
in anticipation of national or global mandates. A new HFC scenario is developed based on
current trends in HFC use and current policies in several countries. These current policies
reduce projected emissions in 2050 from the previously calculated 4.0-5.3 $GtCO_2$-eq $yr^{-1}$ to

1.9-3.6 $GtCO_2$-eq $yr^{-1}$. The provisions of the Kigali Amendment are projected to reduce the
emissions further to 0.9-1.0 $GtCO_2$-eq $yr^{-1}$ in 2050. Without current policies, HFCs would be
projected to contribute 0.28-0.44 °C to the global surface warming in 2100, compared to
0.14-0.31 °C with current policies, but without the Kigali Amendment. In contrast, the Kigali
Amendment controls are expected to limit surface warming from HFCs to about 0.04 °C in

55    2100.

Keywords: Montreal Protocol; Kigali Amendment; radiative forcing; temperature; climate



# 1 Introduction

Hydrofluorocarbons (HFCs) are largely used as alternatives for ozone-depleting substances, which are being phased out as a result of the provisions of the Montreal Protocol (UNEP, 2020). Consequently, large percentage increases have been observed in the emissions and atmospheric abundances of many HFCs since the beginning of this century (Montzka and Velders et al., 2018). HFCs do not deplete the ozone layer, but they are potent greenhouse

gases contributing to climate warming. Global projections from 2009 and 2015 anticipated large increases in HFC use and emissions in the absence of regulations (Velders et al., 2015;Velders et al., 2009). Similar increases were suggested by other studies (Purohit et al., 2020;Gschrey et al., 2011;Purohit and Hoglund-Isaksen, 2017). The projected increase in emissions results in an increase in radiative forcing of the climate – leading to a potential

contribution to global average surface warming of 0.3-0.5°C in 2100 (Montzka and Velders et al., 2018;Xu et al., 2013). In the EU, USA, and Japan, regulations to limit the use of HFCs were already in effect before the Kigali controls (EU, 2014, 2006;METI, 2015;US-EPA, 2020). In 2016, the Kigali Amendment to the Montreal Protocol was agreed to and aims to phase down the production and consumption of HFCs globally, starting in 2019 (UNEP,

2020). Following the provisions of this Amendment, HFC emissions were projected to be constrained and the surface warming is expected to be limited to less than 0.1 °C in 2100 (Montzka and Velders et al., 2018).

In 2015, global projections of HFC use, emissions, mixing ratios, and radiative forcing were reported based on observation-based estimates of emissions through 2012 and observations of

mixing ratios up to 2013 (Velders et al., 2015). Since then, HFC emissions inferred from observed mixing ratios have been reported for several countries, such as the United Kingdom up to 2018 (Manning et al., 2021), the United States from 2008 through 2014 (Hu et al., 2015;Hu et al., 2017), China for 2011-2017 (Yao et al., 2019), and India for 2016 (Say et al., 2019). Also, reviews of bottom-up and top-down emission estimates of HFCs have recently

been published by Montzka and Velders et al. (2018) and Flerlage et al. (2021). Historical HFC consumption data for several use sectors in China have been reported by Li et al. (2019) and Fang et al. (2016) and for Chinese room air conditioning (AC) specifically by Liu et al. (2019).

In this paper, we examine how the global 2015 scenario from Velders et al. (2015) compares

with trends in emissions inferred from atmospheric observations since 2013, to investigate if there are indications of reduced use or slower increase in use of HFCs, e.g., from national



regulations or the Kigali Amendment or the anticipation of the Kigali controls. Based on this comparison, updated historical HCFC consumption data reported to UNEP, HFC use derived from data reported by developed countries to the UNFCCC, and HFC use data reported for

China, we develop new HFC projections and discuss differences with the 2015 scenarios. Apart from the new and updated atmospheric observational information, the current projections also differ from the baseline projections of Velders et al. (2015), because they follow the HFC phasedown policies currently in place in the EU, USA, and Japan.

The structure is as follows. First, in Sect. 2.1 to 2.5 the observations and methods used to

infer global emissions are described, historical HFC and HCFC consumption are discussed, HFC use and emission scenarios are presented, and the model to project HFC emissions and the model assumptions are given. Second, in Sect. 3.1 to 3.3 the 2015 baseline scenario is compared with recent trends in global emissions inferred from observations. Third, the results of the new "current policy" scenario and of the impact of the Kigali Amendment on

emissions are presented in Sect. 3.4 and 3.5, followed by the effects of hypothetical zero production and emissions scenarios in Sect. 3.6. Finally, the effects of the new HFC scenarios on future surface warming are discussed in Sect. 3.7, followed by a discussion and conclusions of the results in Sect. 4.

## 2 Methods

**2.1 AGAGE and NOAA observations and inferred emissions**

HFC mixing ratios are measured by the Advanced Global Atmospheric Gases Experiment (AGAGE) (Prinn et al., 2018) and by the National Oceanographic and Atmospheric Administration (NOAA) Global Monitoring Laboratory GML) (Montzka et al., 2015) at various locations around the globe (see Caption Figure 1). The global and annual average

mixing ratios are used here to infer the global total annual emissions for the individual HFCs from 1990 to 2020 using a 1-box model of the atmosphere and constant prescribed HFC lifetimes as presented in WMO (2018) (see Sect. 2.5). The HFCs considered here are HFC-32, HFC-125, HFC-134a, HFC-143a, HFC-152a, HFC-227ea, HFC-236fa, HFC-245fa, HFC-365mfc, and HFC-43-10mee. These HFCs are used in large amounts for refrigeration and air

conditioning applications, foam blowing, as aerosol propellants, in fire suppression systems, and as solvents. HFC-23 is not considered here, since it has only very small intentional uses and its emissions originate mainly as a by-product from the production of HCFC-22 (Stanley





et al., 2020); furthermore, the Kigali Amendment controls HFC-23 in a different manner than
the other HFCs considered here.

### 2.2 Historical HFC consumption


The new scenarios start with reported historical consumption and assumptions for missing
data using the same procedure as described in Velders et al. (2015). Historical consumption is
based on: 1) HFC activity data and emissions per sector reported by developed countries to
the United Nations Framework Convention on Climate Change (UNFCCC, 2021) up to 2018,

2) HCFC consumption data reported by developing countries to United Nations Environment
(UNEP, 2021) up to 2019, and 3) global HFC emissions derived from global average HFC
mixing ratio histories measured by the AGAGE and NOAA networks up to 2020 (see Sect.
2.1).

Annual consumption of individual HFCs per sector is derived from the UNFCCC (2021)

activity data as the change from one year to the next in the amounts of HFCs in stock plus the
annual HFC emissions from manufacturing, stocks, disposal, and recovery. The derived
consumption data, available for developed (Annex 1) countries only, is grouped in five
regions: European Union (EU), USA, Japan, other countries of the Organization for
Economic Co-operation and Development (OECD), and States of the former Soviet

Republics and Yugoslavia. Consumption for 12 separate use sectors is derived for each
region (see Sect. 2.5). The data reported for Japan to the UNFCCC (2021) is augmented with
consumption data for stationary air conditioning (AC) from their National Inventory report
(NIES, 2020). Consumption data for other reporting OECD countries, only available for
Australia, Canada, New Zealand, Norway, Switzerland, Iceland, and Turkey, is multiplied by

a factor of 1.05 (based on the ratio of the populations) to account for non-reporting OECD
countries. Consumption data for States of the former Soviet Republics and Yugoslavia, only
available for Russia, Ukraine, and Kazakhstan, is multiplied by a factor of 1.5 (population
ratio) to account for other countries in this region.

Consumption data reported for developing countries is only available for China as reported

by Li et al. (2019) for 1995-2017. This consumption data takes into account import and
export of HFCs and therefore represents the Chinese national HFC consumption. HFC
consumption in most other developing countries started around 2013 following the first limits
on the use of HCFCs in developing countries mandated by the Montreal Protocol. Some
counties have reported HFC consumption data for 2018 to UNEP under the requirements of



the Kigali Amendment, but since this data is still incomplete it is not used in the scenarios here. For India, consumption data is estimated from emissions inferred for 2016 by Say et al. (2019) and used in the scenarios. For other developing countries, HFC consumption data for 2013 to 2018 from the baseline scenario of Velders et al. (2015) is used here. These regions are Asian countries other than China and India, Middle and Southern Africa, Latin America,

and Middle East and Northern Africa. HFC-134a consumption data for mobile air conditioning in developing countries (except China) is estimated from the number of cars in use, the average charge and lifetime of cars (see Velders et al. (2009) and Velders et al. (2015)).

From the UNFCCC (2021) data, emissions factors, i.e. the annual emissions as a fraction of

the banks, are derived per sector for the five developed countries/regions. The HFC consumption data from 1990 to 2018 are used in the aforementioned 1-box model of the global atmosphere, in combination with these emission factors and WMO (2018) lifetimes, to calculate annual emissions and mixing ratios. The consumption data is then increased or decreased (by applying a fixed scaling factor to each sector and region) so that the calculated

emissions match the emissions inferred from observations (see Sect. 2.1). This adjustment is applied since uncertainty in emissions derived from observed HFC mixing ratios are small (Montzka and Velders et al., 2018) and assumed to be smaller than emissions estimated from reported consumption data. The 2018 emissions from these calculations, which are used as a starting point for our projections, are therefore consistent with emissions derived from

observed atmospheric mixing ratios.

For some HFCs there is a large mismatch between total calculated emissions from reported or estimated consumption and emissions inferred from observed mixing ratios. This results from no reporting, or underreporting, of HFC consumption by developed countries to the UNFCCC and missing information from developing countries (Flerlage et al., 2021). To

obtain a good starting point for the scenarios, the consumption of these HFCs is scaled so that the calculated emissions match the emissions inferred from observations. Large mismatches are found for HFC-152a (extruded polystyrene (XPS) foam and aerosol use in the USA), HFC-227ea (fire protection and aerosol use in the USA), HFC-236fa (fire protection use in China), HFC-245fa (polyurethane (PUR) foam in the USA), HFC-365mfc (solvent use in the

EU), and HFC-41-10mee (solvent use in the EU and the USA).





### 2.3 "Current policy" scenario for future emissions

Our scenario that is consistent with existing regional and national policies, excluding the
Kigali Amendment, starts with the derived consumption totals from both developed and
developing countries during 1990 to 2017. The total consumption in each year is adjusted so

that the calculated emission of a specific HFC agrees with the emission derived from the
observed atmospheric mixing ratios, using fixed emissions factors. Projections through 2050
(and 2100) are based on assumptions for growth in demand for HFCs and HCFCs (see
below), the phaseout of HCFCs following the provisions of the Montreal Protocol and
assumptions on how much this demand is met by HFCs or not-in-kind alternatives. The

procedure is similar to that of Velders et al. (2015). The baseline scenario of Velders et al.
(2015) assumed unabated growth in HFC use and emissions. Since 2015, regulations to limit
the use of HFCs are in place in the EU (EU, 2006, 2014), the USA (US-EPA, 2020), and
Japan (METI, 2015) and reductions in HFC consumption are observed in several sectors (see
Sect. 3). Therefore, the scenario presented here takes into account the reductions in HFC use

from these regional and national regulations and is referred to as the "current policy"
scenario. This scenario includes legislation adopted by parliaments in countries and
implemented in national regulations as of the end of 2020 and does not include the provisions
of the Kigali Amendment.

The demand for HCFCs and HFCs in developed countries in the various applications is

assumed to be saturated in 2018 and the demand after 2018 therefore changes proportional to
the projected growth or decline in population in each of these five countries or world regions
(see Sect. 2.2). This could underestimate the projected demand for HFCs considering, for
example, the recent push in the EU for heat pumps to replace oil and gas for space heating.
The demand in developing countries is assumed to be the sum of the demand for applications

that already use HFCs and the demand for applications in which HCFCs are currently or
soon-to-be phased out and replaced in part by HFCs; it is assumed to change proportional to
the growth or decline in Gross Domestic Product (GDP) for each of the six countries or world
regions (see Sect. 2.2). The HCFC consumption from 1989 to 2019 reported to UNEP (2021)
is the starting point for determining HCFC demand. The GDP and population data is taken

from the Shared Socioeconomic Pathway (SSP) projections (O'Neill et al., 2012). The
"current policy" scenario consists of an upper range scenario following the GDP and
population in the SSP5 pathway and a lower range scenario following the SSP3 pathway.





In the "current policy" scenario, similar as in the 2015 baseline scenario, market saturation is assumed in developing countries when the demand reaches a level such that the sum of HFC

and HCFC consumption per capita in that country or region reaches a level of the maximum consumption per capita of the developed countries. This saturation level is reached in different years for the different groups of sectors (domestic refrigeration, industrial and commercial refrigeration, stationary AC, mobile AC, foams, and other sectors). After saturation the consumption in each sector changes proportionally to population projections in

a region.

The "current policy" scenario includes the policies in place to limit the use of HFCs, such as the EU mobile air conditioning directive (EU, 2006), the EU revised f-gas directive (EU, 2014), the American Innovation and Manufacturing act (US-EPA, 2020), and regulations in Japan (METI, 2015). This scenario also includes a phaseout of HFC-134a used for mobile air

conditioning following recent trends in several countries. In the EU the use of fluorinated gases with a GWP greater than 150 has been prohibited for new vehicles since 2017. Currently, the vast majority of new vehicles sold in the EU, the USA, and Japan are not equipped with HFC-134a, but with the low-GWP alternative hydrofluoroolefin HFO-1234yf (Taddonio, 2021). HFC-134a may still be used for recharging the AC systems in existing

vehicles. In the "current policy" scenario the consumption of HFC-134a for mobile AC is therefore reduced linearly from 2018 to 2030 in the EU, USA, Japan, and other OECD countries. The amount of HFC consumption in excess of the limits of the current policies is assumed to be replaced by low-GWP alternative substances of alternative technologies.

In Sect. 3.2 we will show that the global emissions of HFC-143a inferred from observed

mixing ratios is much lower than previously projected (Velders et al., 2015) and more or less constant from 2016 to 2019. This shows that the consumption of R-404A (the blend in which HFC-143a is mainly used in) for industrial and commercial refrigeration has been lower in recent years than previously projected. Therefore, for the upper range of the "current policy" scenario the consumption of HFCs for industrial and commercial refrigeration is constant at

the 2018 level for all developing countries. For the lower range of the "current policy" scenario, future consumption for these applications is still proportional to the growth in GDP in developing countries, allowing for the transition and use of other HFCs in this application.

Following the provisions of the Montreal Protocol (UNEP, 2020), the consumption of HCFCs in developing countries is reduced stepwise with a full phaseout in 2040. In the baseline

scenario the demand for HCFCs is then replaced by HFCs and not-in-kind replacements. We





use here the same replacement schedule as by Velders et al. (2009) and Velders et al. (2015): HCFC-22 is replaced by R-410A and R-404A (90%) and low-GWP or not-in-kind alternatives (10%); HCFC-141b and HCFC-142b are replaced by HFC-245fa and HFC-365mfc (50%) and by low-GWP alternatives (50%).

**2.4 Kigali Amendment scenario**

The HFC phasedown schedule of the 2016 Kigali Amendment is applied to the consumption in the "current policy" scenario for a Kigali scenario assuming adoption by all countries of the world. The phasedown is applied to the total $CO_2$-equivalent ($CO_2$-eq) HFC consumption using 100-yr GWPs to scale down the consumption in all sectors of a country or region by the

same factor. The amount of HFC consumption in the "current policy" scenario in excess of the limits of the Kigali Amendment is assumed to be replaced by low-GWP alternative substances or alternative technologies.

**2.5 Box model**

The historical and projected HFC consumption is used in a global 1-box model to calculate

the amounts present in equipment (the bank), the emissions, mixing ratios, and radiative forcing for each of the 10 HFCs, 12 use sectors, and 11 regions from 1990 to 2050 (or 2100) (see Velders and Daniel (2014) and Velders et al. (2015)). The emissions are calculated as a fraction of the bank as estimated from the UNFCCC (2021) data, see Table S1 (Supplement). In the box model a fixed atmospheric lifetime is used. The atmospheric lifetimes, the global

warming potentials (100 year time horizon), and radiative efficiencies are from WMO (2018). The HFC observations from both the AGAGE and NOAA/GML networks are used in the comparison of the inferred emissions with the emissions of the 2015 baseline scenario (Sect. 3.1 and 3.2). As a starting point for the new scenarios, only the observations from the AGAGE network are used (Sect. 3.4 to 3.6).

In the model, 12 separate use sectors taken are considered, 1) industrial refrigeration, 2) commercial (open compressor, hermetically sealed compressor), 3) transport refrigeration, 4) domestic refrigeration, 5) stationary AC, (6) mobile AC, 7) extruded polystyrene foams (XPS), 8) polyurethane foams (PUR), 9) open cell foams, 10) aerosol products, 11) fire extinguishing systems, and 12) solvents.



The contribution of the HFCs to the global average surface warming is calculated from the emissions and atmospheric mole fractions derived from 1990 to 2100 using the low complexity carbon cycle-climate model MAGICC6 (Meinshausen et al., 2011a). This model has been calibrated with output from complex coupled atmosphere-ocean general circulation models and applied in several simulations (Meinshausen et al., 2011b).

## 3  Results

### 3.1  Emissions baseline scenario compared with recent trends

The 2015 baseline scenario from Velders et al. (2015) was based on reported HFC data for developed countries up to 2011, reported HCFC consumption data up to 2013, and observed HFC mixing ratios up to 2013. The global total $CO_2$-eq emissions from the 2015 baseline

scenario are compared with emissions inferred from HFC observations from the AGAGE and NOAA/GML networks from 2010 to 2020 (Figure 1). The inferred emission global totals are below the baseline emissions by about 20% in the period 2017-2019.

The baseline emissions of the HFCs with the largest global use, i.e., HFC-32, HFC-125, HFC-134a, and HFC-143a, are compared, individually, with the emissions inferred from

observed mixing ratios (Figure 2). A comparison of the emissions of the other HFCs are shown in Figure S1 and a comparison of the mixing ratios in Figure S2 (see Supplement). The baseline emission projections of HFC-134a are in good agreement with the inferred emissions. The inferred emissions of HFC-32 and HFC-125 are lower than the baseline emissions, by about 19% and 25%, respectively, averaged over 2017-2019. The largest

difference is seen for HFC-143a; the inferred emissions are about 40% lower than the baseline emissions for 2017-2019.

The global total inferred emission of HFC-143a slowly increased through 2016, but are more or less constant from 2016 to 2019 (Figure 2). HFC-143a is used mainly (for 90% or more in our scenario) in the blend R-404A (52% HFC-143a, 44% HFC-125, 4% HFC-134a) for

industrial and commercial refrigeration applications (UNFCCC, 2021;UNEP, 2019). In the 2015 baseline scenario it was assumed that the global consumption of R-404A would increase following the growing demand for refrigeration applications, mainly in developing countries, and the phaseout of HCFC-22 under the Montreal Protocol. The fact that the global emissions of HFC-143a are significantly below the 2015 baseline scenario and constant in

recent years, shows that R-404A is used in smaller amounts than expected to replace HCFC-





22 for refrigeration applications. Other alternatives are likely used in larger amounts in this sector than expected previously (Velders et al., 2015). In Europe R-404A has already been replaced by an HFC blend (R-452A) without HFC-143a for new trucks (transport refrigeration), while propane and $CO_2$ are also being used in commercial refrigeration

applications (UNEP, 2019).

### 3.2 HFC-143a consumption trends

Until about 2013, the majority (more than 80%) of the HFC-143a emissions came from use in developed countries in the 2015 baseline scenario, estimated from the reported consumption data. In that scenario the consumption in developing countries increased by about a factor of

3.5 from 2013 to 2019, while that in developed countries increased by only 27% in the same period. Consequently, the 2015 scenario anticipated that 2019 emissions of HFC-143a would originate from developed and developing countries in equal amounts.

The EU and USA have the largest reported use (consumption) of HFC-143a of all developed (Annex 1) countries (UNFCCC, 2021). The use in the EU has decreased by about 60% in

2017 compared to 2010. This decrease is ahead of the ban for HFCs with a GWP larger of 2500 or more for commercial refrigeration applications, in effect since 2020 (EU, 2014). In the USA the use of HFC-143a increased by 14% in 2017 compared to 2010, while small increases are also seen in other developed countries. In sum, the total Annex 1 use (consumption) decreased by about 8% in 2017 compared to 2010 (Figure 3).

The largest growth in HFC-143a in the baseline scenario was projected for China, with an increase of a factor of six from 2010 to 2017 (Velders et al., 2015). This projected increase was based on the cap and initial phaseout of HCFC-22 and large projected economic growth in China. According to Li et al. (2019) the consumption of HFC-143a in China also increased by a factor of six from 2010 to 2017, but the absolute values are much lower than in the

scenario. The consumption for China in the baseline scenario was based on Zhang and Wang (2014) and Fang et al. (2016) and is about ten times larger than that reported by Li et al. (2019) for 2005-2013; a 2013 consumption is 0.8 kt in Li et al. (2019) versus 6.4 kt in Fang et al. (2016). The lower reported consumption of HFC-143a from Li et al. (2019) takes into account export of HFCs. Information on export of HFCs, which is significant for HFC-143a,

was not available for the studies from Zhang and Wang (2014) and Fang et al. (2016) (personal communication Jianxin Hu).





Based on the decreased use of HFC-143a in the EU, limited increases in other developed countries, and lower reported consumption in China, the "2015 baseline" was adjusted to test if the observed trend in emissions can be matched with a simple adjustment. In this "Reduced

HFC production ICR" scenario the use of HFCs for industrial and commercial refrigeration (ICR) in developed countries was reduced following the UNFCCC reported reduction in HFC-143a from 2013 to 2020, while the use in developing countries was held constant at the 2013 level. With these adjustments to the baseline scenario in the "Reduced HFC production ICR" scenario, emissions of HFC-143a and also of HFC-125 (also part of the blend R-404A)

are close to the emissions inferred from observations (Figure 2). With this single adjustment the $CO_2$-eq emissions of all HFCs closely follow the emissions inferred from observations (Figure 1). This adjustment was directly included in the "current policy" scenario since it is based on the historical UNFCCC reported data.

### 3.3 Reported HCFC and HFC consumption

The total reported HCFC consumption in both developed and developing countries is decreasing (Figure 3). In 2019, the consumption in developed countries was about 98% below the 1990s peak, while the consumption in developing countries was about 60% lower than the peak in 2012. The reported HCFC phaseout in developed countries is already virtually complete in line with the assumption in the "current policy" scenario discussed here

(Sect. 2.3). For both groups of countries, the total HCFC consumption was below the limits set by the Montreal Protocol for all years. This potentially provided additional pressure on demand for HFCs to replace HCFCs and compounding the demand associated with the required HCFC phaseout schedule.

The global HFC consumption derived here from the reported UNFCCC (2021) activity data

and emissions shows strong increases for HFC-32 and HFC-125 (Figure 3), prior to scaling to match observation derived emissions, albeit less than projected in the 2015 baseline scenario. Both compounds are predominantly used in the blend R-410A in stationary AC (about 90% of HFC-32 and 63% of HFC-125 in 2017), leading to the conclusion that the use of HFCs in stationary ACs is increasing strongly in developed countries.

The consumption of HFC-134a increased in developed countries until about 2010 after which it started to decrease (Figure 3). This HFC is used in many applications of which mobile AC is the largest; about 43% of all HFC-134a consumption in developed countries was in this sector in 2017. The decrease since 2010 is mainly from less use in mobile AC in the EU and



USA. The consumption for mobile AC in the EU and USA stabilized between 2000 and 2010
and then decreased, probably as a result of smaller charges and less frequent recharging of
existing systems (as a result of reduced leakage rates). Adding to this slowdown was the
introduction of HFO-1234yf for mobile AC around 2012 (Vollmer et al., 2015), reducing the
use of HFC-134a for this applications further in the EU and USA.

The trend in consumption of HFC-143a is discussed in Sect. 3.2.

**3.4  Current policy scenario**

The $CO_2$-eq emissions and radiative forcing of the "current policy" scenario are compared
with the 2015 baseline scenario for the period up to 2050 (Figure 4). The difference in
scenarios from 2010 to 2020 arises from the difference between the projections of the 2015
scenario and the inferred observations included in the "current policy" scenario. Both
scenarios use the emissions inferred from observed mixing ratios as a starting point. For the
2015 baseline scenario, observational-based emissions were available up to 2012 and for the
"current policy" scenario they were available through the year 2019. The 2050 emissions in
"current policy" scenario are 1.6-2.1 $GtCO_2$-eq yr$^{-1}$ lower than in the baseline scenario (1.9-
3.6 versus 4.0-5.3 $GtCO_2$-eq yr$^{-1}$). This difference comes from the different assumptions in
the projections, mainly the lower projected consumption of HFC-125 and HFC-143a for
industrial and commercial refrigeration in developed and developing countries (see Sect. 3.2).
Smaller contributions come from the transition from HFC-134a to HFO-1234yf in mobile AC
in developed countries, and the implementation of other provisions of the current regional
and national HFC regulations in the EU, the USA, and Japan (see Sect. 2.3).

The effect of the regulations in the EU, the USA, and Japan, including the changes in mobile
AC, are estimated to reduce the global emissions by 0.4-0.5 $GtCO_2$-eq yr$^{-1}$ in 2050, relative to
the "current policy" scenario (not shown).

The effects of the regulations are also evident in the emissions projections of HFC-32, -125, -
134a, and -143a in Figure 5. See Figure S3 for the other HFCs in the Supplement. The lower
emissions in HFC-125 and HFC-143a in the "current policy" scenario compared to the 2015
baseline scenario (Figure 5) contribute most to the difference in $CO_2$-eq emissions (Figure 4)
due to the large GWPs of 3450 and 5080 (WMO, 2018), respectively. These differences in
emissions translate into a difference in 2050 radiative forcing of 0.08-0.09 W m$^{-2}$ (0.13-0.18
W m$^{-2}$ in the "current policy" scenario versus 0.22-0.25 W m$^{-2}$ in the baseline scenario).


The contributions of the different world regions and application sectors to the $CO_2$-eq emissions and radiative forcing of the upper range of the "current policy" scenario are shown in Figure 6. While the contributions to the emissions and radiative forcing in developed countries decrease after about 2020 and 2040, respectively, the contributions from developing countries continue to grow in the absence of the provisions of the Kigali Amendment. The

largest contributions are from China (34% in $CO_2$-eq emissions and 40% in radiative forcing for the upper range of the "current policy" scenario for 2050), followed by India (19% and 13%, respectively). Considering the use sectors, the largest contributions for the upper range of the "current policy" scenario are from ICR and stationary AC applications with 41% and 37%, respectively of the total in 2050. For the lower range of the "current policy" scenario

the contributions from ICR applications are strongly reduced and stationary AC applications have the largest contributions to both emissions and radiative forcing, with 56% and 47%, respectively, in 2050 (not shown).

### 3.5  Kigali Amendment scenario

The provisions of the 2016 Kigali Amendment are projected to reduce the 2050 HFC

emissions from 1.9-3.6 $GtCO_2$-eq $yr^{-1}$ in the "current policy" scenario to 0.9-1.0 $GtCO_2$-eq $yr^{-1}$ (Figure 4). This is slightly lower than projected relative to the 2015 baseline scenario, because of a lower Kigali baseline levels (average consumption of 2020-2022 or 2024-2026) for developing countries. The corresponding radiative forcing in the "current policy" scenario is reduced from 0.13-0.18 W $m^{-2}$ to 0.09-0.10 W $m^{-2}$ in 2050 by the provisions of the Kigali

Amendment.

### 3.6  Zero production and emissions scenario

In two hypothetical scenarios that incorporate a cessation in global production or emissions of HFCs in 2023, the emissions and radiative forcing is further reduced (Figure 4). With a cessation in production in 2023 the radiative forcing drops to about 0.03 W $m^{-2}$ by 2050,

while if all emissions (from new production and from banks) cease in 2023 it is reduced to about 0.01 W $m^{-2}$ by 2050, compared to 0.09-0.10 W $m^{-2}$ considering the provisions of the Kigali Amendment.



### 3.7  Surface temperature contributions by HFCs

Radiative forcing of greenhouse gases contributes to global surface warming, changes in
atmospheric circulation, and other effects. The contribution of emissions and radiative
forcing of HFCs (Figure 4) to the surface warming is calculated using the MAGICC6 model
(Meinshausen et al., 2011a) and shown in Figure 7. For this calculation the scenarios are
extended to 2100 based on the same assumptions as used for the period 2020-2050. In the
"current policy" scenario the HFCs are projected to contribute 0.14-0.31 °C to the global
average surface warming in 2100, compared to 0.28-0.44 °C in the 2015 baseline scenario
(Velders et al., 2015;Montzka and Velders et al., 2018). With the provisions of the Kigali
Amendment the surface warming of the HFCs drops to about 0.04 °C in 2100. For
comparison, all greenhouse gases are projected to contribute 1.4-4.8 °C to the surface
warming by the end of the 21[st] century following the RCP6 and RCP8.5 scenarios (IPCC,
2013). In hypothetical scenarios with a cessation in global production or emissions of HFCs
in 2023 the contribution to the surface warming is reduced to virtually zero (0.01 °C and
0.004 °C in 2100, respectively).

## 4  Discussion and conclusions

We analyzed trends in observations of atmospheric abundances of ten different HFCs from
1990 to 2020 and the emissions inferred from these observations and compared them with
previous projections. Total $CO_2$-eq HFC emissions inferred from observations continue to
increase through 2019, but are about 20% lower than previously projected for 2017-2019.
The main reason is lower global inferred emissions of HFC-143a during 2012-2019. This
HFC is mainly used in the blend R-404A in ICR applications. Data reported to the UNFCCC
also show lower than projected consumption of HFC-143a with a reduction in use in the EU
and a stabilization in the USA and Japan. Consumption data reported for China also shows a
reduced use of HFC-143a (Li et al., 2019). These lower emissions and reduced consumption
cannot be linked directly to the provisions of the Kigali Amendment, since that only came
into force in January 2019. It could indicate that companies transitioned away from higher
GWP HFCs, HFC-143a in particular, for ICR applications prior to national or global
regulation mandates. For example, in the EU, the use of any HFC with a GWP of 2500 or
more, such as HFC-143a (GWP 5080), is banned from use in ICR applications since January
2020, but companies appear to have switched to lower-GWP or not-in-kind alternatives



before this date. This could also explain the observed emissions reduction of HFC-125 which

is used together with HFC-143a in R-404A.

The HFC emissions reported by Annex 1 (developed) countries to the UNFCCC are roughly constant throughout the period 2014-2018 and account for about half of global total emissions inferred from observations during these years. This gap is predominantly not the result of underreporting by Annex 1 countries (at least not from the EU and the US, for which

atmospheric measurement based emissions estimates are in general agreement with UNFCCC reporting), but instead is associated with substantial emissions from non-Annex 1 countries (Montzka and Velders et al., 2018).

New HFC scenario projections are developed based on the trends in the use of HFCs in developed and developing countries, current policies in effect in the EU, the USA and Japan,

and emissions inferred from observed abundances as constraints until 2019. In this "current policy" scenario, the 2050 HFC emissions of 1.9-3.6 $GtCO_2$-eq yr$^{-1}$ are substantially lower than in a baseline scenario derived without any control measures or anticipation of control measures of 4.0-5.3 $GtCO_2$-eq yr$^{-1}$. The provisions of the Kigali Amendment in addition to the current policies are projected to reduce the emissions further to 0.9-1.0 $GtCO_2$-eq yr$^{-1}$ in

2050, which is very similar to total HFC emissions in 2019 of about 0.8 $GtCO_2$-eq yr$^{-1}$.

Global emissions of all fluorinated gases have recently been estimated by Purohit et al. (2020) using projections of activity data in various sectors and taking into account national and regional emission controls in place as of 2016 when the Kigali Amendment was adopted. Their baseline 2050 emissions are about 4.3 $GtCO_2$-eq yr$^{-1}$, which is consistent with our 2015

baseline, but higher than our "current policy" scenario that includes the consideration of additional control measures in some countries.

Li et al. (2019) reported projections of HFC use and emissions for China. Their 2050 business-as-usual HFC emissions are about 0.9 $GtCO_2$-eq yr$^{-1}$, which agrees well with our "current policy" (e.g., not including the Kigali Amendment) estimate for China of 0.9-1.3

$GtCO_2$-eq yr$^{-1}$. In our scenario we use the Chinese historical HFC consumption from 1995 to 2017 from Li et al. (2019) as a starting point, but with somewhat different assumptions with respect to growth rates and market saturation. Li et al. (2019) estimated total Chinese HFCs emissions in 2050, considering compliance with the Kigali Amendment, of about 0.20 $GtCO_2$-eq yr$^{-1}$ which is slightly lower than our estimate of about 0.43 $GtCO_2$-eq yr$^{-1}$. Liu et

al. (2019) reported projections specifically for room AC in China and estimated 2050


emissions of 0.13-0.26 $GtCO_2$-eq $yr^{-1}$ for this use sector. This number cannot be directly compared with our scenario since we have one sector of stationary AC, which consists of both residential and commercial AC.

Projected mixing ratios, radiative forcing, and globally averaged temperature changes are
calculated from the projected HFC emissions. The 2050 radiative forcing is 0.13-0.18 W $m^{-2}$ in the "current policies" scenario and drops to 0.08-0.09 W $m^{-2}$ when the additional Kigali Amendment controls are considered. In the current policies scenario, the HFCs are projected to contribute 0.14-0.31 °C to the global surface warming in 2100, compared to 0.28-0.44 °C without policies. Following the Kigali Amendment, the surface warming of HFCs is reduced
to about 0.05 °C in 2050 and 0.04 °C in 2100. In a hypothetical scenario with a full phaseout of HFCs production and consumption in 2023 the contribution is reduced to about 0.01 °C in 2100.

Projected $CO_2$-eq emissions, radiative forcing, and climate warming calculated here are only from the direct effects of projected emissions of HFCs. There are also indirect emissions of
greenhouse gases associated with the production and use of HFCs in various applications. For example, AC systems use electricity to operate and, depending on the manner in which the electricity is generated, significant emissions of $CO_2$ can occur. This and other indirect effects arising from the use of HFCs and/or alternative substances and technologies need to be considered to estimate the complete effect of HFCs and alternatives and the Kigali
Amendment on climate. For example, switching to superefficient low-GWP alternatives in room AC systems and cooling equipment in general has both direct and indirect climate benefits (Shah et al., 2015, 2019). Purohit et al. (2020) estimated significant electricity savings from a global phasedown of HFCs in addition to the saving in GWP-weighted emissions from HFCs directly.

In conclusion, the current observed trends in developed and developing countries and policies in several developed countries reduce the projected global average surface warming attributed to HFC emissions (excluding HFC-23) by about 0.14 °C (from 0.28-0.44 to 0.14-0.31 °C) in 2100 compared to earlier estimates that didn't include the updated data and new controls. A further reduction of 0.10-0.26 °C (to about 0.04 °C) is possible in 2100 with the
global implementation and compliance with the provisions of the Kigali Amendment.



**Acknowledgements.** We thank Jianxin Hu (University of Beijing) for information about use of HFC in China and Kristen Taddonio (Institute for Governance & Sustainable Development) for information of HFC use for mobile AC. The five AGAGE stations from

which ambient measurements were used here are supported by the National Aeronautics and Space Administration (NASA) (grants NNX16AC98G to MIT, NNX16AC97G and NNX16AC96G to SIO, and preceding grants). Support also comes from the UK Department for Business, Energy & Industrial Strategy (BEIS, contract 1537/06/2018 to the University of Bristol) for Mace Head, the National Oceanic and Atmospheric Administration (NOAA,

contract 1305M319CNRMJ0028 to the University of Bristol) for Ragged Point, and the Commonwealth Scientific and Industrial Research Organization (CSIRO) and the Bureau of Meteorology (Australia) for Cape Grim. NOAA measurements of HFCs benefited from the technical assistance of C. Siso, B. Hall, B. Miller, M. Crotwell, personnel at remote sampling stations, and funding in part from the NOAA Climate Program Office's AC4 program.


**Author contributions**. GJMV designed the research, collected and analyzed the consumption data, and performed the calculations. JSD, SAM and MR assisted with the research design. Measurement data were collected by SAM, IV, MR, PBK, JM, SOD, RGP, RFW, and DY. GJMV wrote the article, with contributions from all co-authors.






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





**Figures**

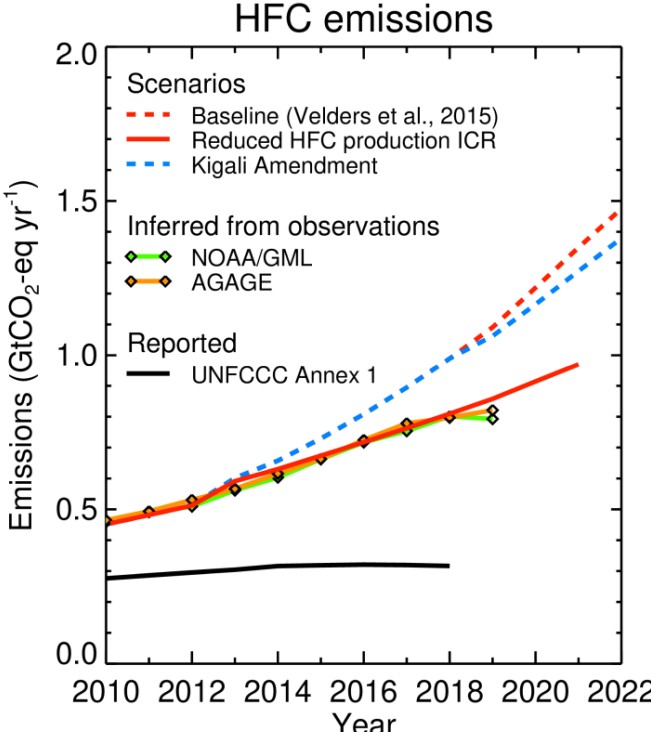

Figure 1: Global total HFC emissions ($GtCO_2$-eq yr$^{-1}$) from the 2015 baseline scenario, the
2016 Kigali Amendment scenario, and inferred from observed mixing ratios from the
AGAGE and NOAA/GML networks (see methods). The solid line shows an adjusted
scenario with reduced HFC consumption for industrial and commercial refrigeration (ICR)
from 2013 onward. The curves contain the contributions from all HFC emissions, except
HFC-23. The scenarios were constrained by the emissions inferred from observed mixing
ratios up to 2013. Also shown are the emissions reported to the UNFCCC by developed
(Annex 1) countries.





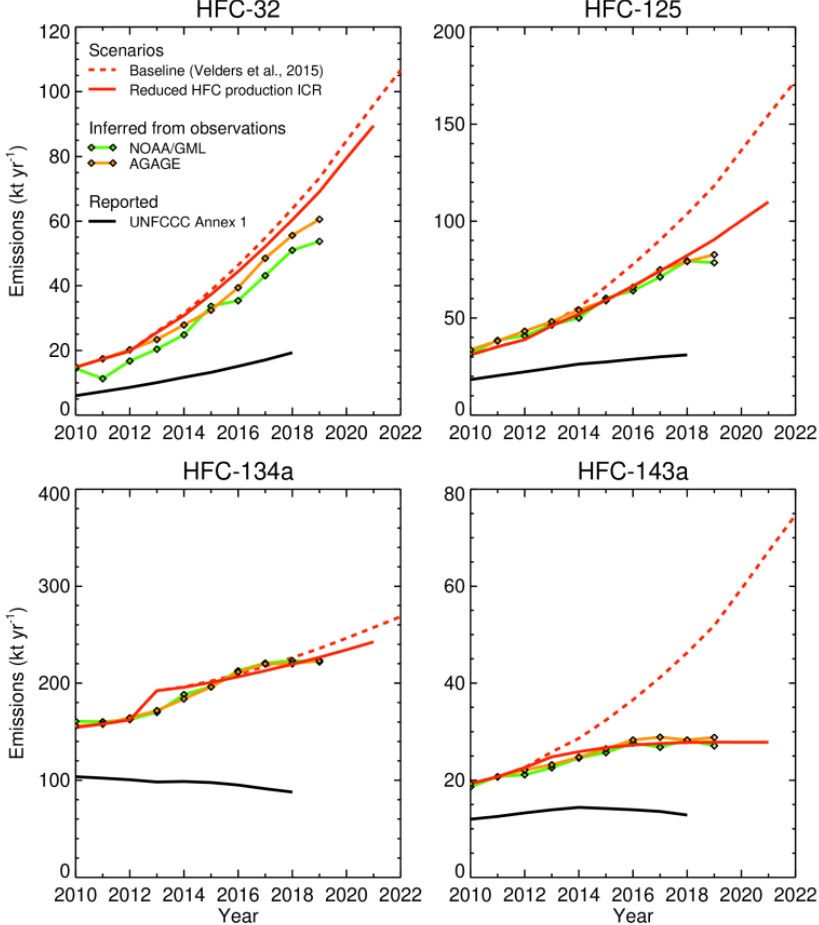

Figure 2: Global total HFC emissions (kt yr$^{-1}$) from the 2015 baseline compared with
emissions inferred from observed mixing ratios from the AGAGE and NOAA/GML
networks. The solid line shows an adjusted scenario with reduced HFC production for
industrial and commercial refrigeration (ICR) from 2013 onward. Also shown are the
emissions reported to the UNFCCC by Annex 1 countries. The scenarios were constrained by
the emissions inferred from observed mixing ratios up to 2013. The AGAGE data is from the
observations at Mace Head (Ireland), Trinidad Head (Barbados), Ragged Point (Barbados),
Cape Matatula (American Samoa), Cape Grim (Tasmania). The NOAA data is from the
stations at Alert (Canada), Barrow (Alaska), Niwot Ridge (Colorado), Mauna Loa (Hawaii),
Kumukahi (Hawaii), American Samoa, Cape Grim (Tasmania), and South Pole.






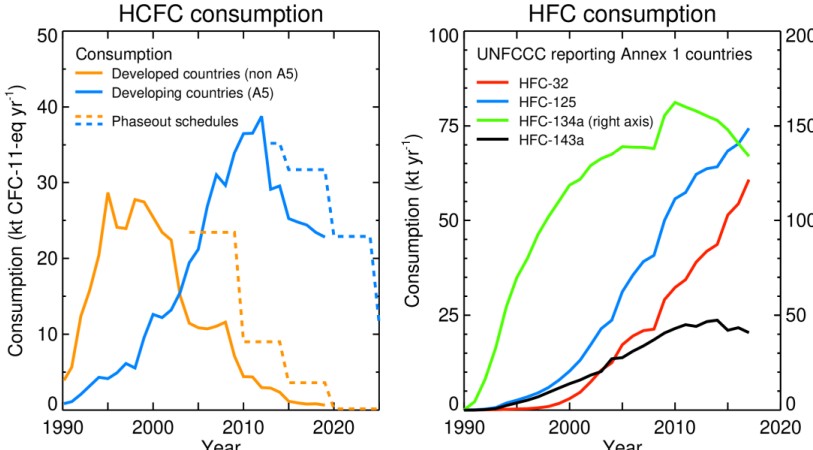

Figure 3: Left panel: HCFC consumption (kt CFC-11-eq yr$^{-1}$) in developed (non A5) and developing (A5) countries as reported to UNEP (2021) and the Montreal Protocol phaseout schedules. Right panel: HFC consumption (kt y$^{-1}$) in developed countries (Annex 1 countries)
derived from the data reported to the UNFCCC (2021).



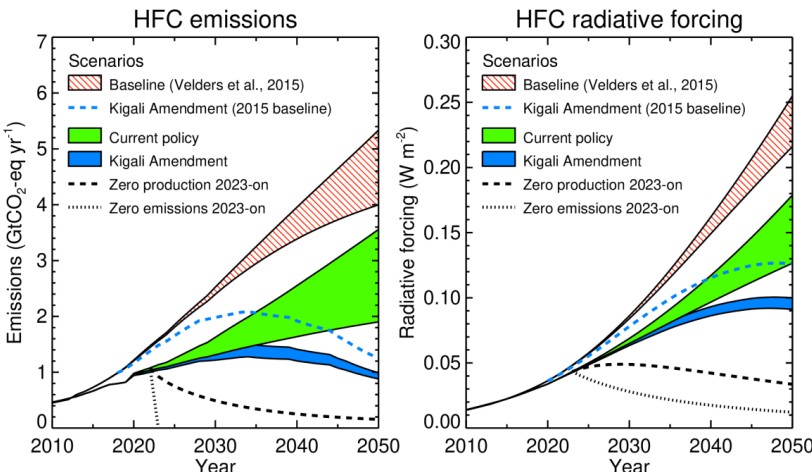

Figure 4: Global total HFC emissions ($GtCO_2$-eq yr$^{-1}$) (left panel) and radiative forcing (right panel) from the 2015 baseline scenario and the "current policy" scenario. The bands represent the upper and lower ranges of these scenarios. Also shown are scenarios that follow the phasedown schedules of the 2016 Kigali Amendment, based on the 2015 baseline, and the "current policy" scenario. Also shown are hypothetical scenarios in which the global HFC production ceases in 2023 or the global HFC emissions (from new production and from banks) cease in 2023. The curves contain the contributions of all HFCs except HFC-23.


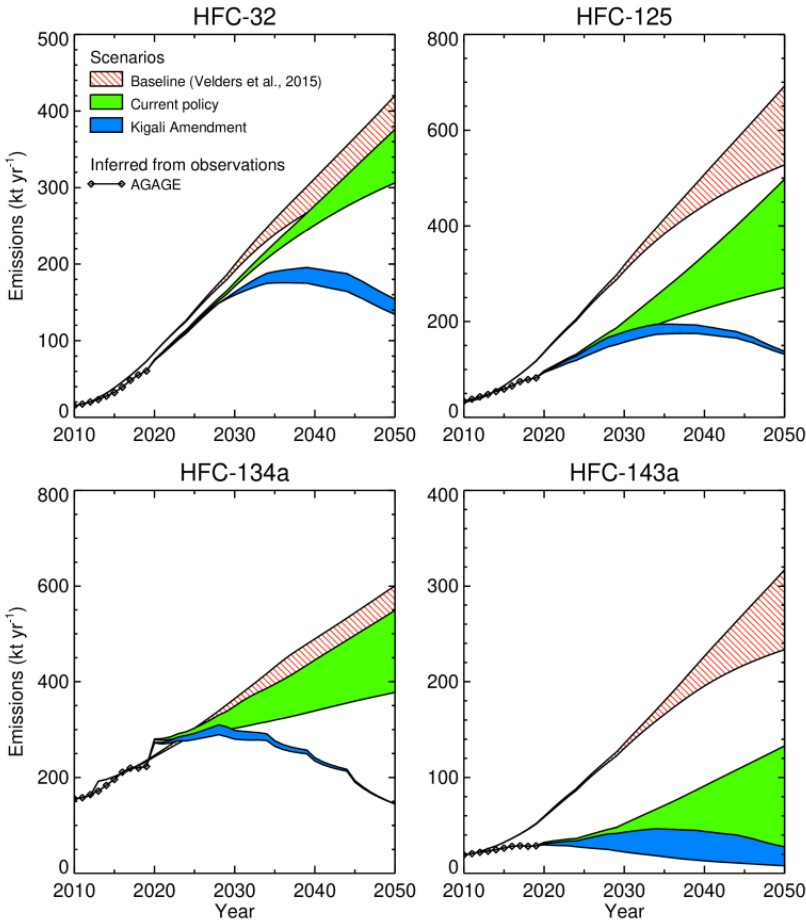

Figure 5: Global total emissions (kt yr$^{-1}$) of HFC-32, HFC-125, HFC-134a, and HFC-143a from the 2015 baseline scenario, the "current policy" scenario, and the Kigali Amendment scenario. The bands represent the upper and lower ranges of these scenarios. Also shown the
emissions inferred from observed mixing ratios from the AGAGE network.



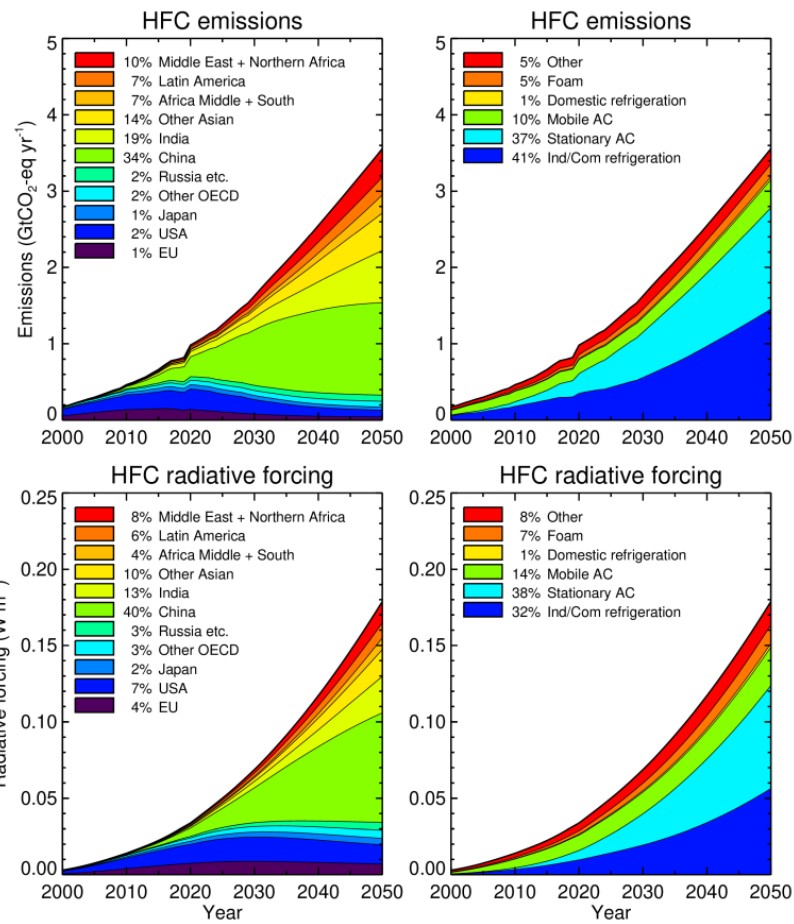

Figure 6: Global total HFC emissions ($GtCO_2$-eq yr$^{-1}$) (top panels) and radiative forcing
(bottom panels) from the upper range of the "current policy" scenario. Shown are the
contributions from the different regions (left) and different sectors (right). The percentages
refer to the relative contributions of the $CO_2$-eq emissions and radiative forcing in the upper
range in 2050.

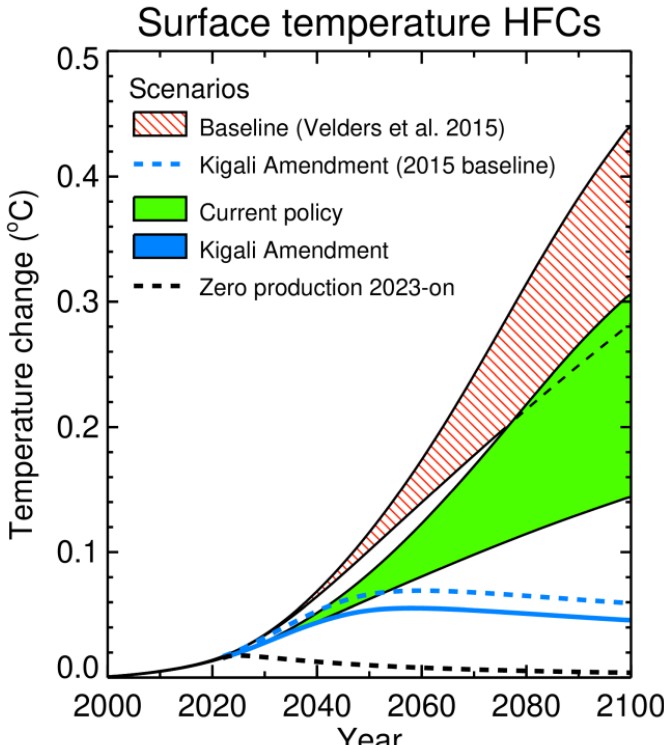

Figure 7: Contribution of HFCs to the global average surface warming for the 2015 baseline
scenario without measures on HFC consumption and the "current policy" scenario. The bands
represent the upper and lower ranges of these scenarios. Also shown are the effects of the
phasedown of HFCs following controls of the Kigali Amendment and a hypothetical scenario
assuming that the global production of emissions of HFCs would cease in 2023. The surface
temperature change is calculated using the MAGICC6 model (Meinshausen et al., 2011a).
The curves contain the contributions of all HFCs, except HFC-23.