# Peer review of "Projections of hydrofluorocarbon (HFC) emissions and the resulting global warming based on recent trends in observed abundances and current policies"

_Atmospheric Chemistry and Physics, 2021_

## Referee Comment (RC1)

**Title: Projections of hydrofluorocarbon (HFC) emissions and the resulting global warming based on recent trends in observed abundances and current policies**

**Journal: Atmospheric Chemistry and Physics**

**Manuscript Ref: acp-2021-1070**

**General comments**

The authors have analyzed the trends in HFC emissions inferred from observations of atmospheric abundances and compare them with previous projections. The results indicate that as compared to the previous projections HFC consumption is lower in industrial and commercial refrigeration sector that is also supported by data reported by the developed countries and China. As a next step, a new HFC scenario is developed based on current trends in HFC use and current policies in several countries. Finally, impact on global surface warming is estimated in the new and several alternative scenarios.

The novel aspect of this study is that the authors have developed a new scenario for HFC emissions inferred from observations of atmospheric abundances and analyzed the warming impacts attributed to HFC emissions in the alternative scenarios as well. In my opinion, with an improved presentation this paper is acceptable for publication in ACP. I have few major and other minor comments to improve my understanding of the topic.

**Major comments**

1. P3, L64-65: You have mentioned that "HFCs do not deplete the ozone layer, but they are potent greenhouse gases contributing to climate warming." Please note that HFCs do not destroy ozone directly, but they can indirectly lead ozone depletion through radiative impacts.
2. P4, L68-77: Based on this discussion what I understand is that the full compliance with the Kigali Amendment to the Montreal Protocol will avoid approximately 0.2 to 0.4°C additional warming by the end of this century. Please confirm!
3. P7, L196-199: There were regulations to limit the use of HFCs in place apart from the EU, USA and Japan. For e.g., the Swiss Regulation on Substances Stable in the Atmosphere, from December 2013, bans on many HFC uses, including larger air-conditioning, commercial and industrial refrigeration. Similarly, in Norway, a GWP-weighted excise duty on the import and production of HFCs and PFCs (including HFC-134a in mobile air-conditioning systems in imported cars), was introduced in 2003 and has steadily been increased since, its rate in 2015 was NOK354 (about €39) per tonne $CO_2$-eq. Similar regulations were in place in Australia (carbon tax), New Zealand (HFC levy) etc.
4. P7, L208: As I understand, heat pumps (ground, water and air) are not included in this analysis or merged with other sectors. Please confirm?
5. P9, L260-262: There are several alternative options available for a given sector. For example, $NH_3$ ($GWP_{100} = 0$), HFOs ($GWP_{100} = <1$), $CO_2$ ($GWP_{100} = 1$) to HFC-32 ($GWP_{100} = 677$). How do you settle on which low-GWP alternative will be selected to replace the amount of HFC consumption in the "current policy" scenario more than the limits of the Kigali Amendment?
6. P9, L268 (Table S1): How reliable are the reported leakage rates from the banks, particularly Russia? The leakage rates are lower in Russia as compared to EU. I do not see the leakage rates for the end-of-life emissions. As I understand, except EU and few industrialized countries not all the parties to the Montreal Protocol have good practices in place for the recovery of the refrigerants after the end-of life of the cooling equipment. How do you estimate the end-of-life emissions?
7. P9, L269: Why the GWP values provided in the latest IPCC/AR6 are not considered in this study?

8.  P12, L367-368: HFC-410A (R-410A) is a zeotropic mixture of 50% HFC-32 and 50% HFC-125. It would be nice if you could rephrase "…(about 90% of HFC-32 and 63% of HFC-125 in 2017)…" indicating that this is not the composition of HFC-410A but the total consumption of HFC-32 and HFC-125 in stationary air-conditioning sector. One of the reasons of the high share of HFC-32 is that the split AC manufacturers are switching from high-GWP HFC-410A to low-GWP HFC-32 due to the existing national/regional and global (i.e., Kigali Amendment) regulations.

9.  P14, L408: Why transport refrigeration is not shown in Figure 6? Is it included with others? This needs to be shown separately from the other small industrial sectors.

10. P14, L426: To be honest, I do not understand the purpose of these two hypothetical scenarios - Zero production and emissions scenario. What message they convey? What are the assumptions considered in this scenario? I think in the zero-production scenario you have considered the refrigerant bank in the equipment that will remain until the lifetime of the cooling system. Instead of zero production and emissions scenario what seems more realistic is an accelerated Kigali Amendment scenario. Also, avoid the repetition of the same results (in Section 3.7 and Section 4) on the contribution to the surface warming in these hypothetical scenarios.

**Minor comments**

1.  P2, L44: Kigali controls or obligations/requirements?
2.  P4, L93, Provide abbr. HCFC?
3.  P4, L93, Provide abbr. UNEP?
4.  P4, L94, Provide abbr. UNFCCC?
5.  P4, L113, Please correct - …Global Monitoring Laboratory (GML)…
6.  P8, L233, Please correct - …hydrofluoroolefin (i.e., HFO-1234yf)… as several other hydrofluoroolefin such as HFO-1234ze, HFO-1336mzz-E, HFO-1336mzz-Z, etc. are also being used in the market.
7.  P12, L372-373: Provide reference – "about 43% of all HFC-134a consumption in developed countries was in this sector in 2017."
8.  P15, L462: Please rephrase …($GWP_{100}$ = 5080)…

---

## Author Comment (AC1)

**Response to the comments of the reviewers**

We thank the reviewers for their insightful and constructive comments. The time they spent on our manuscript is very much appreciated and has helped make the manuscript scientifically stronger and clearer. We have revised the manuscript based on the suggestions made by the reviewers.

**Reviewer #RC1**

Title: Projections of hydrofluorocarbon (HFC) emissions and the resulting global warming based on recent trends in observed abundances and current policies

**Journal: Atmospheric Chemistry and Physics**

**Manuscript Ref: acp-2021-1070**

**General comments**

The authors have analyzed the trends in HFC emissions inferred from observations of atmospheric abundances and compare them with previous projections. The results indicate that as compared to the previous projections HFC consumption is lower in industrial and commercial refrigeration sector that is also supported by data reported by the developed countries and China. As a next step, a new HFC scenario is developed based on current trends in HFC use and current policies in several countries. Finally, impact on global surface warming is estimated in the new and several alternative scenarios.

The novel aspect of this study is that the authors have developed a new scenario for HFC emissions inferred from observations of atmospheric abundances and analyzed the warming impacts attributed to HFC emissions in the alternative scenarios as well. In my opinion, with an improved presentation this paper is acceptable for publication in ACP. I have few major and other minor comments to improve my understanding of the topic.

**Major comments**

1. P3, L64-65: You have mentioned that "HFCs do not deplete the ozone layer, but they are potent greenhouse gases contributing to climate warming." Please note that HFCs do not destroy ozone directly, but they can indirectly lead ozone depletion through radiative impacts.

**Response**: Agreed. This has been shown by Hurwitz et al. (2015). The sentence has been changed and a reference to Hurwitz et al. (2015) has been added.

2. P4, L68-77: Based on this discussion what I understand is that the full compliance with the Kigali Amendment to the Montreal Protocol will avoid approximately 0.2 to 0.4°C additional warming by the end of this century. Please confirm!

**Response**: That is correct. Previous publications estimated that HFCs contribute 0.3 to 0.5C to the surface warming in 2100. With the Kigali amendment this will be reduced to less than 0.1C. Therefore, the Kigali amendment is projected to reduce warming by 0.2 to 0.4C by 2100. A sentence has been added to Sect 1 stating this explicitly.

3. P7, L196-199: There were regulations to limit the use of HFCs in place apart from the EU, USA and Japan. For e.g., the Swiss Regulation on Substances Stable in the Atmosphere, from

December 2013, bans on many HFC uses, including larger air-conditioning, commercial and industrial refrigeration. Similarly, in Norway, a GWP-weighted excise duty on the import and production of HFCs and PFCs (including HFC-134a in mobile air-conditioning systems in imported cars), was introduced in 2003 and has steadily been increased since, its rate in 2015 was NOK354 (about €39) per tonne CO2-eq. Similar regulations were in place in Australia (carbon tax), New Zealand (HFC levy) etc.

**Response**: You are correct. There are also regulations to reduce the use of HFCs in place ion other countries. I have added "and several other countries' to Section 1 and 2.3.

4. P7, L208: As I understand, heat pumps (ground, water and air) are not included in this analysis or merged with other sectors. Please confirm?

**Response**: Heat pumps are not addressed explicitly in the scenario, but as part of Industrial and Commercial Refrigeration (ICR). That is, the current use of heat pumps is projected to grow according to the general growth in ICR. Extra growth in heat pumps to replace space heating is not included and could represent an underestimation of the scenarios. This is addressed in Sect. 2.3.

5. P9, L260-262: There are several alternative options available for a given sector. For example, NH3 (GWP100 = 0), HFOs (GWP100 = <1), CO2 (GWP100 = 1) to HFC-32 (GWP100 = 677). How do you settle on which low-GWP alternative will be selected to replace the amount of HFC consumption in the "current policy" scenario more than the limits of the Kigali Amendment?

**Response**: Indeed, several alternatives are available to replace the high-GWP HFCs. Following the provisions of the Kigali Amendment the total GWP-weighted consumption and production must be reduced. In the Kigali Amendment scenario we scale down the GWP-weighted consumption in a country to be in agreement with the phasedown schedule. The reduced demand is replaced by an 'artificial' alternative with a GWP100 of 10. A sentence has been added to Sect. 2.4 explaining this better.

6. P9, L268 (Table S1): How reliable are the reported leakage rates from the banks, particularly Russia? The leakage rates are lower in Russia as compared to EU. I do not see the leakage rates for the end-of-life emissions. As I understand, except EU and few industrialized countries not all the parties to the Montreal Protocol have good practices in place for the recovery of the refrigerants after the end-of life of the cooling equipment. How do you estimate the end-of life emissions?

**Response**: The leakage rates for developed countries are the overall emission factors (as fraction from the banks) as derived from the UNFCCC data. This data is reported by the individual countries themselves. Countries have to report their emissions according to the IPCC guidelines. Although the methods and data is frequently reviewed by IPCC experts, the quality of the emission factors is not well known. Due to lack of other information, we have used the data as reported by the individual countries. Text has been added explaining this.

7. P9, L269: Why the GWP values provided in the latest IPCC/AR6 are not considered in this study?

**Response**: The GWP100 from IPCC/AR6 are about 10% larger than those we used from WMO(2018), in part because of the larger  $CO_2$  mixing ratio used as reference for the GWP calculation. In the Kigali Amendment the GWP from AR4 have been included. The GWPs from WMO(2018) have been used for the calculations of the baseline scenario in Velders et al. (2015). For consistency we chose to use the same values here. The choice of

GWP has no effect on the scenarios and surface temperature calculation, only on the absolute values of the CO2-eq emissions presented in the text and figures. A sentence has been added to Sect. 2.5 mentioning the effect of the use of larger GWPs from AR6.

8. P12, L367-368: HFC-410A (R-410A) is a zeotropic mixture of 50% HFC-32 and 50% HFC-125. It would be nice if you could rephrase "...(about 90% of HFC-32 and 63% of HFC-125 in 2017)..." indicating that this is not the composition of HFC-410A but the total consumption of HFC-32 and HFC-125 in stationary air-conditioning sector. One of the reasons of the high share of HFC-32 is that the split AC manufacturers are switching from high-GWP HFC-410A to low-GWP HFC-32 due to the existing national/regional and global (i.e., Kigali Amendment) regulations.

**Response**: Agreed. The current sentence is confusing. It has been rephrased.

9. P14, L408: Why transport refrigeration is not shown in Figure 6? Is it included with others? This needs to be shown separately from the other small industrial sectors.

**Response**: Transport refrigeration (not mobile AC) is part of Industrial and Commercial Refrigeration (ICR). For developed countries, there is sufficient data (from the UNFCCC) to model it as a separate sector. For developing countries the data is limited and therefore part of the larger sector ICR. Text has been added to the caption of Figure 6 explaining this.

10. P14, L426: To be honest, I do not understand the purpose of these two hypothetical scenarios

- Zero production and emissions scenario. What message they convey? What are the assumptions considered in this scenario? I think in the zero-production scenario you have considered the refrigerant bank in the equipment that will remain until the lifetime of the cooling system. Instead of zero production and emissions scenario what seems more realistic is an accelerated Kigali Amendment scenario. Also, avoid the repetition of the same results (in Section 3.7 and Section 4) on the contribution to the surface warming in these hypothetical scenarios.

**Response**: Sentence added to address this concern. These hypothetical scenarios are not achievable, but allow one to identify the extreme lower limits on future radiative forcing from HFCs so, we believe, are useful to discern. This is now mentioned in the manuscript.

**Minor comments**

- 1. P2, L44: Kigali controls or obligations/requirements? **Response**: 'controls' changed to 'provisions.
- 2. P4, L93, Provide abbr. HCFC? **Response**: Done
- 3. P4, L93, Provide abbr. UNEP? **Response**: Done
- 4. P4, L94, Provide abbr. UNFCCC? **Response**: Done
- 5. P4, L113, Please correct ...Global Monitoring Laboratory (GML)...

**Response: Done**

6. P8, L233, Please correct - ...hydrofluoroolefin (i.e., HFO-1234yf)... as several other hydrofluoroolefin such as HFO-1234ze, HFO-1336mzz-E, HFO-1336mzz-Z, etc. are also being used in the market.

**Response**: This sentence discusses the use of HFO-1234yf in mobile AC. Other HFOs are being used for other applications. A sentence has been added mentioning this.

7. P12, L372-373: Provide reference – "about 43% of all HFC-134a consumption in developed countries was in this sector in 2017."

**Response**: Reference to UNFCCC has been added to the sentence.

8. P15, L462: Please rephrase ...(GWP100 = 5080)... **Response**: Done

**Reviewer** #RC2**

**Projections of hydrofluorocarbon (HFC) emissions and the resulting global warming based on recent trends in observed abundances and current policies** Guus J.M. Velders, John S. Daniel, Stephen A. Montzka, Isaac Vimont, Matthew Rigby, Paul B. Krummel, Jens Muhle, Simon O'Doherty, Ronald G. Prinn, Ray F. Weiss, and Dickon Young

The paper develops a scientifically relevant hydrofluorocarbon (HFC) emission scenario consistent with current observations and policies, and compares it with several other scenarios, especially with the one reported by the main author in 2015. The paper infers that under the newly developed scenario, the HFCs contribution to global warming is significantly smaller than expected by the author in 2015, and explains this with political measures and technical developments implemented in the meantime. The effects of complying with the Kigali Amendment are also estimated and discussed. Changes in HFCF consumption over time are shown for developed and developing countries from 1990 to 2020 and related to replacing HFC substances. Time series of emissions under a variety of scenarios and two different observation based emission estimates are shown for four HFC and the total of the most important HFC species. HFC emission scenarios according to world region and with respect to sectorial emissions are plotted, together with their respective radiative forcings. Various HFC scenarios and their contribution to global average surface temperature 2000-2100 are presented.

This paper is valuable and of interest, also for policy makers and general public, as it presents a detailed description on the HFC emission situation and projection and its relevance for climate change.

Major points to revise:

- The paper needs a discussion on the scientific limitations of the used methods.
  **Response:** The model and methods used in this study have been described in detail in previous publications (Velders and Daniel, 2014; Velders et al., 2015, and previous work) and presented in the UNEP/WMO scientific assessment of ozone depletion, 2018. This paper focusses how the previous projections, based on the same model, compare with recent observations of HFCs and on comparing different scenarios. Complex atmospheric chemistry transport models could be used for this, but a box model has been shown to be adequate for comparing scenarios.
  Text has been added to Sect. 1, 2.5, and 4 mentioning uncertainties in the model and the purpose of the paper.
- Error discussion, preferably error bars should be provided with given results, without which interpretation of (significance of) results is sometimes difficult.

**Response**: The focus of our study is on comparing different scenarios with respect to emissions, radiative forcing, and surface temperature, not on estimating the best absolute values. An important factor for the overall uncertainty are the assumption for the projected HFC consumption. We therefore present ranges for the projections; upper and lower ranges based on projected growth the demand for HFCs. Uncertainties also originate from, e.g., the boxmodel (lifetime of the HFCs), the emission factors, radiative properties of the HFCs, and the temperature calculation. These uncertainties are less important when comparing scenarios. Text has been added to Sect. 1 and 4 addressing this.

• The number of observations, and the representativity of stations for the purpose of calculating global averages, and deriving emissions, as well as uncertainty of this value should be discussed for all analysed substances. In the Acknowledgements, it can be found that possibly only 5 stations are used? What robustness can be achieved for the different substances? A plot of the global average concentrations (possibly with variability estimate or confidence interval for the mean value) for all 10 discussed substances would be illustrative. This would allow the lines 449-450 to stay, otherwise line 449-450 has to be removed, as observations and trends are not shown in this paper. Also "ten" is misleading, discussed in this paper are 4 substances and the sum of ten.

**Response**: The stations with observations used in the analyses are mentioned in the caption of Figure 2 (now moved to the caption of Figure 1). For the AGAGE network this are five stations and for the NOAA network eight stations. All stations are located at various latitudes across the globe. Because of the relatively long atmospheric lifetime of the HFCs they are distributed more or less homogeneously across the globe. Global and annual average mixing ratios can therefore be calculated from the five of eight stations with high confidence. For the AGAGE stations this is done by a 12-box model; 1-sigma uncertainties of a few percent are estimated for the global average mixing ratios for the four major HFCs and similar results are derived for global concentrations and emissions by NOAA using a different suite of sites and a different modeling approach (as demonstrated in the figures; see also Montzka and Velders et al., 2018). In the main paper only the four main HFCs are shown and discussed, but in the Supplement also the data from the six other HFCs are shown. In Figure S2 all the global average observed mixing ratios of all 10 HFCs are shown. The uncertainties are added to the caption of Figure S2 and text has been added to Sect. 2.1 explaining this.

• The reference to the 1-box model (line 116) is insufficient to understand how the emissions were derived from concentrations, especially as later there seems to be a split according to regions, sectors and developed/developing countries - how does this fit with a 1-box model? Also in section 2.5, the box model is not described in such a way the reader can understand whether the application of MAGICC6 is fit for purpose. Any error bars coming out of MAGICC6? Are (line281) the mole fractions (used for the purpose of this paper ?) calculated by MAGICCS from the emissions given by the scenarios? How do the mole fraction outputs from MAGICC6 compare to the AGAGE/NOAA observations? Or are in fact the averaged observed values used as an input to MAGICC6?

**Response**: The description in sect. 2.1 is indeed rather concise. Details of how emissions are calculated are described at length (with all the formulas) in Velders et al. (2015). Text has been added to section 2.1 and 2.5 presenting more information on the calculations. The MAGICC6 model uses the global HFC emissions as input. The HFC lifetimes, GWP100 and radiative forcing data used in MAGICC6 are identical to the data used in the scenario presented in the paper. The mixing ratios calculated by MAGICC6 are therefore also the same as we calculate. As our scenario, the MAGICC6 model does not calculate uncertainties for the emissions and surface temperatures. The MAGICC6 model has been validated and the results compared with output from the RCP scenarios. Clearly, uncertainties are associated with our scenarios and with the calculated temperatures. These uncertainties originate from, e.g., the boxmodel (lifetime of the HFCs), the emission factors, radiative properties of the HFCs, and the temperature calculation. Uncertainties are also associated with the

scenarios itself. We therefore present upper and lower ranges for the scenarios, originating from the SSP scenarios. In our study, the focus is on comparing the effects of different scenarios, not on the absolute values. The uncertainties in the model and parameters are therefore less important. An extensive uncertainty analyses of ozone depleting substances using the same box model has been reported earlier (Velders and Daniel, 2014). Text has been added to Sect. 2.5 and 4 mentioning the uncertainties and purpose of the study.

Minor points:

Several scenarios are used. They need to be clearly explained, and preferably, memorisable named. A table would help the reader to keep track which ones are compared. My understanding:

Line 33: Scenario A

Line 37: Scenario B1 and B2 (using AGAGE and NOAA/GML, respectively) – why are not data sources from AGAGE and NOAA/GML used together (that would be scenario B3 Line 47: Scenario C1 "current trends and current policies"

Line 48: Scenario C2 "current policies" – here it is unclear whether actually same as C1 is meant, or whether the omitting of "current trends" marks the difference to C1. Line 50: Scenario D (Kigali amendments)

Line 51: Scenario E1 "without current policies" ... is this the same as scenario A? E1 is surely also without Kigali amendments, line 53 seems to suggest otherwise Line52: Scenario E2: "current policies, but without Kigali Amendment" is this the same as

C1?

**Response**: We agree that the scenarios and their names could be confusing. The names are changes somewhat and labels are added: Baseline V-2015, Current Policy Kigali Independent (K-I), Kigali Amendment (KA-2015 and KA-2022). This changes have been implemented in the text and all figures.

In the text as well as in the figures, it is sometimes, but not always clear which scenario exactly is meant, e.g.,

Line 420: not sure which scenario of the above is meant, or a new one.

**Response**: We agree that the scenarios and their names could be confusing. The names are changes somewhat and labels are added: Baseline V-2015, Current Policy Kigali Independent (K-I), Kigali Amendment (KA-2015 and KA-2022). This changes have been implemented in the text and all figures.

- Location of observations are not shown (as promised in line 114). **Response**: Well spotted. The stations were mentioned in the caption of Figure 2. This is now moved to Figure 1.
- References like "see Sect. 2.1" at line 170 or "see Sect. 3" at line 198/119 are not helpful, also line 117, 132 etc, as no additional information can be found there. **Response**: You are correct. The reference to Sect. 2.1 has been changed to Sect. 3.1 and Reference to Sect. 3 has been changed to Sect 3.3. The reference to Sect. 2.5 on line 117 is relevant, since that sections described the box model calculations. A reference to Sect. 3.1 has been added to line 132.
- Reference to figures: include "right" or "left" for clarity, e.g., line 329, 350, 356, 365, 371.

**Response**: For clarity, 'left' of 'right' pane has been added to the figure reference in several locations.

- Line 509: check wording "direct effect of projected emissions" probably better: "effects of direct emissions" (direct/indirect refers to emissions here, not to effects)
  Response: Agreed. The sentence is changed.
- Supplement: Superscripts are indicating the data source. Superscript of a line seems to be valid for al numbers of the line, except when the number has its own superscript (this would be a reasonable rule). There is also a column with a superscript does the column-superscript predominate the line superscript? Easier would be to apply the same reasonable rule.

**Response**: The superscript could indeed be confusing. I changed it somewhat and explain the emission factor that used for the developing countries in the note below the table.

I would like to encourage the authors to tackle these major and minor points (from a reader perspective), as they should be not too much effort from the authors' perspective and will improve readability and usefulness of this potentially nice paper.

---

## Author Response (AR2)

Comment editor April 5, 2022

**Comments to the author**:
dear Guus,

thank you for the revised manuscript, which I am happy to accept for publication in ACP. I only have one very minor comment. In the abstract (l. 32 of the track chnaged manuscript) you mention that the " projections showed ."... I believe that this may be misleading and would suggest to rephrase to something like " Emissions were projected to ... ", as I believe that it is misleading to say that a projection shows something.

best regards, Andreas

Response, April 10, 2022

The sentence has been changed to "In 2015, large increases were projected in HFC use and emissions in this century in the absence of regulations, …"